# Association between Gut Microbiota and SARS-CoV-2 Infection and Vaccine Immunogenicity

**DOI:** 10.3390/microorganisms11020452

**Published:** 2023-02-10

**Authors:** Ho Yu Ng, Wai K. Leung, Ka Shing Cheung

**Affiliations:** 1School of Clinical Medicine, The University of Hong Kong, Hong Kong 999077, China; 2Department of Medicine, School of Clinical Medicine, Queen Mary Hospital, The University of Hong Kong, Hong Kong 999077, China

**Keywords:** gut microbiota, gut dysbiosis, COVID-19, COVID-19 vaccine, vaccine immunogenicity

## Abstract

Gut microbiota is increasingly recognized to play a pivotal role in various human physiological functions and diseases. Amidst the COVID-19 pandemic, research has suggested that dysbiosis of the gut microbiota is also involved in the development and severity of COVID-19 symptoms by regulating SARS-CoV-2 entry and modulating inflammation. Previous studies have also suggested that gut microbiota and their metabolites could have immunomodulatory effects on vaccine immunogenicity, including influenza vaccines and oral rotavirus vaccines. In light of these observations, it is possible that gut microbiota plays a role in influencing the immune responses to COVID-19 vaccinations via similar mechanisms including effects of lipopolysaccharides, flagellin, peptidoglycan, and short-chain fatty acids. In this review, we give an overview of the current understanding on the role of the gut microbiota in COVID-19 manifestations and vaccine immunogenicity. We then discuss the limitations of currently published studies on the associations between gut microbiota and COVID-19 vaccine outcomes. Future research directions shall be focused on the development of microbiota-based interventions on improving immune response to SARS-CoV-2 infection and vaccinations.

## 1. Introduction

Since its emergence in late 2019, the coronavirus disease 2019 (COVID-19) pandemic has affected all healthcare systems and societies across the world, accumulating over 700 million confirmed cases and resulting in over six million deaths worldwide as of early February 2023. An interesting feature of COVID-19 is its ability to cause symptoms outside of the respiratory tract. A meta-analysis revealed that a pooled prevalence of 17.6% of patients had gastrointestinal symptoms, which included diarrhea, nausea, vomiting, anorexia, and abdominal pain or discomfort [1]. The occurrence of gastrointestinal symptoms has further been suggested to be linked to a more severe disease course in another meta-analyses [2]. Therefore, there is evidence suggesting the interplay between COVID-19 and the gut.

Moreover, the gut microbiota have been increasingly recognized to play a role in COVID-19 pathophysiology. The gut microbiota consist of as many as 100 trillion micro-organisms [3] and the collective metagenome can be as many as 150 times more than the human genome [4]. Its composition is also highly variable among individuals from different age groups, geographical location, and lifestyle habits [5]. As such, the gut microbiota are often regarded as the “forgotten organ” of the human body and they potentially influence various metabolic activities and body functions. Recent studies have reported that compared to normal individuals, COVID-19 patients have altered gut microbiota composition and dysbiosis, which in turn can influence COVID-19 disease course and severity through gut barrier dysfunction, altered ACE2 expression, and influencing the gut–lung axis [6]. The gut microbiota have also been reported to be able to modulate immune response to various vaccines [7]. Emerging evidence has also shown that the potential role of gut microbiota in modulating COVID-19 vaccine immunogenicity, and that their variation may also be part of the reason why COVID-19 vaccine immunogenicity may differ substantially among different individuals.

In this review, we will give an overview on the interaction between the gut microbiota and the disease course of COVID-19. We will also highlight current knowledge on the role of the gut microbiota in vaccine immunogenicity including the COVID-19 vaccine.

## 2. Relationship between Gut Microbiota and SARS-CoV-2 Infection

As it has been increasingly shown that gut microbiota plays a pivotal role in the human immune system, the association between the gut microbiota and COVID-19 has been extensively studied during the pandemic. A considerable number of cross-sectional studies performed on animals and humans alike demonstrated that gut microbiota dysbiosis was observed during SARS-CoV-2 infection, though whether this was the cause or the effect of SARS-CoV-2 infection remained not fully understood. Nonetheless, gut microbiota dysbiosis appears to modulate COVID-19 severity and clinical outcomes, while in turn SARS-CoV-2 infection may induce alterations in the gut microbiota. As such, gut microbiota dysbiosis is hypothesized to have a bidirectional relationship with COVID-19 and its outcomes.

### 2.1. Gut Microbiota Dysbiosis Associated with Susceptibility to SARS-CoV-2 Infection and Disease Severity

While evidence that directly implicates the gut microbiota in affecting a person’s susceptibility to SARS-CoV-2 is still currently lacking, it has been suggested that the gut microbiota dysbiosis can increase the risk of SARS-CoV-2 infection by modulating the expression of the viral entry receptor angiotensin-converting enzyme 2 (ACE2) in the gut and by regulating B cells and T cells [8]. Animal and human studies, however, have shown that gut microbiota dysbiosis could be associated with more severe clinical outcomes. In a study conducted on healthy hamsters, several taxa of the gut microbiota were strongly correlated with inflammatory responses to SARS-CoV-2 infection. Positive correlations with lung histological scores and inflammatory cytokines were observed in *Christensenellaceae*, *Desulfovibrioaceae*, *Flavobacteriaceae*, and *Peptococcaceae* families, while negative correlations were seen in *Butyricicoccaceae* and *Ruminococcaceae* [9]. Similarly, in another study conducted on obese NASH hamsters, *Blautia* and *Peptococcus* were positively correlated with pro-inflammatory or pro-fibrotic profiles, whereas in lean hamsters *Gordonibacter* and *Ileibacterium* were negatively correlated with inflammatory profiles [10].

For human subjects, a study from Hong Kong showed that the COVID-19 patient cohort was found to have significant enrichment in *Ruminococcus gnavus*, *Ruminococcus torques*, and *Bacteroides dorei*, but lack *Bifidobacterium adolescentis*, *Faecalibacterium prausnitzii*, and *Eubacterium rectale* [11]. After adjusting for antibiotic use and patients’ age, *F. prausnitzii* and *Bifidobacterium bifidum* were found to have a significant negative correlation with COVID-19 severity. In addition, *B. adolescentis*, *E. rectale*, and *F. prausnitzii*, which were known to have immunomodulatory effects in the human gastrointestinal system, were negatively correlated with various immune markers. The depletion of these species may have contributed to overaggressive inflammation and even cytokine storms seen in severe COVID-19 cases. Another study from Japan showed similar findings [12]. In this study, the gut microbes enriched in the COVID-19 patient cohort, which included *R. torques*, were positively correlated with cytokines that were enriched during COVID-19, including those that were implicated with increased disease severity and cytokine storms. In contrast, gut microbes that were depleted in the COVID-19 cohort, which included *B. adolescentis* and *E. rectale*, were correlated with cytokines that were reduced during COVID-19, including CCL20 which was important for regulatory T cell migration. This suggested that the gut microbiota was involved in cytokine metabolism, which was in turn linked to inflammation and disease severity. In another Hong Kong study, 23 bacterial taxa in the baseline gut microbiome were found to be significantly associated with the severity of COVID-19, most of which were *Firmicutes* [13]. Again, *F. prausnitzii* were most negatively correlated with COVID-19 severity. A study conducted in Germany also found that the relative abundance of *Faecalibacterium* and *Roseburia* was lower in severe or critical COVID-19 cases [14]. Therefore, the gut microbiota and COVID-19 are very likely to have a dynamic relationship which can potentially form a vicious cycle with lasting effects.

The potential associations between the gut microbiota and COVID-19 severity were also indirectly implicated by studies that investigated the associations between the use of proton pump inhibitors (PPIs) and COVID-19 severity. These studies showed that PPI usage may increase the risk of COVID-19 positivity [15] and severity [16]. In an American study, there was a dose–response relationship in which those who took PPIs twice daily had a higher risk of being COVID-19-positive compared to just a single daily dose [15]. Lower-dose PPI use was also associated with lower odds of developing gastrointestinal symptoms of COVID-19 in individuals tested positive for COVID-19. In a Korean study, however, propensity score matching found that PPI use was not associated with COVID-19 positivity, but was associated with 79% increased risk of severe symptoms of COVID-19 [16]. PPIs alter the gut microbiome with significant increase in relative abundance of *Enterococcus*, *Streptococcus*, *Staphylococcus*, and the potentially pathogenic *E. coli* [17]. PPIs may also inhibit the activities of immune cells [18], increase the risk of enteric infections by suppressing gastric acid secretion [19], and alter immunomodulatory and anti-inflammatory effects [20]. In the context of the COVID-19 pandemic, the above effects of PPIs could have resulted in higher SARS-CoV-2 viral load in the GI tract, which contributed to more severe clinical outcomes [16].

### 2.2. COVID-19-Induced Gut Microbiome Alterations

COVID-19 could potentially induce alterations in the gut microbiota. Animal studies have demonstrated that SARS-CoV-2 infection in macaques was able to induce changes in gut microbiota and metabolome, which peaked at 10–13 days post infection [21]. In the hamster study mentioned previously, SARS-CoV-2 infection was characterized by the enrichment of deleterious bacterial taxa, including *Enterobacteriaceae* and *Desulfovibrionaceae*, as well as decreased relative abundance of several members of *Ruminococcaceae* and *Lachnospiraceae* families which included bacteria known to produce short-chain fatty acids (SCFAs) [9]. In obese hamsters, such SARS-CoV-2-induced changes in the gut microbiota could persist even longer [10]. This evidence suggested that the gut microbiota may be subjected to perturbations brought by body response to SARS-CoV-2.

A number of cross-sectional studies on human subjects have reported gut microbiota alterations in COVID-19 cases. (Table 1) In adult COVID-19 patients, there was decreased bacterial diversity in the fecal microbiome. Compared to healthy individuals, COVID-19 patients were observed to have reduced abundance of SCFA-producing bacteria, specifically *Faecalibacterium*, *Eubacterium*, *Coprococcus*, *Ruminococcus*, *Lachnospira*, and *Roseburia*, as well as increased abundance of opportunistic pathogens from Enterbacteriaceae families, specifically *Enterococcus, Rothia* and *Lactobacillus*. Notably, at the genus level, *Bifidobacterium*, *Bacteroides*, *Streptococcus*, and *Enterococcus* were most commonly reported to be enriched, while at the species level, *Bifidobacterium longum* and *Ruminococcus torques* were each reported in two separate studies (Table A1). On the other hand, the genera *Faecalibacterium*, *Eubacterium*, *Coprococcus*, *Bifidobacterium*, and *Clostridium* were most commonly reported to be depleted, while at the species level *Eubacterium hallii* and *Eubacterium rectale* were each reported in two separate studies (Table A1). In children with COVID-19, the genera *Akkermansia* and *Bifidobacterium* were most commonly found to be depleted (Table A1). One study found increased abundance of *Faecalibacterium, Fusobacterium*, and *Neisseria*, as well as decreased abundance of *Bifidobacterium*, *Blautia*, *Granulicatella*, and *Prevotella* in infected children [22]. In another smaller study, the alteration of gut microbiome was predominated by Pseudomonas, and such alteration could sustain for up to almost two months. [23] Another study conducted in asymptomatic children reported decreased abundance of *Bifidobacterium bifidum* and *Akkermansia muciniphila* in SARS-CoV-2-positive stool samples [24]. These two species were linked to protection against inflammation in previous studies. In particular, multisystem inflammatory syndrome in children (MIS-C), which was associated with COVID-19, was commonly found to have enriched level of *Clostridium* (Table A1). Studies have also found that these children had decreased *Bifidobacterium* [22] and *Firmicutes* including *Faecalibacterium prausnitzii* [25]. Of note, these studies were only observational in nature and were not able to show that the gut microbiota alterations were the result of SARS-CoV-2 infection as the baseline gut microbiome before SARS-CoV-2 infection was not profiled. Another interesting observation was that the gut microbiome composition in COVID-19 appeared to be distinct from that in patients with influenza [26,27] or viral pneumonia [13], which suggested that SARS-CoV-2 may have unique effects on the gut microbiome compared with other respiratory viruses [6].

Gut microbiota are also thought to be involved in the development of post-acute COVID-19 syndrome (PACS), or “long COVID”, which is associated with persistent respiratory, cardiovascular, neuropsychiatric, gastrointestinal, and dermatological symptoms [36,37]. Gut microbiome alterations observed in acute COVID-19 episodes could persist even after clearance of SARS-CoV-2 infection [11,13]. A study conducted in Hong Kong found that at 6 months, the gut microbiota of PACS patients were significantly depleted of *Collinsella aerofaciens*, *F. prausnitzii*, and *Blautia obeum*, as well as enriched with *Ruminococcus gnavus* and *Bacteroides vulgatus* [38]. In contrast, patients without PACS had fewer species altered, and the alterations were able to recover at 6 months. The relative abundance of *Bifidobacterium pseudocatenulatum*, *F. prausnitzii*, *Roseburia inulinivorans*, and *Roseburia hominis*, which were known to be beneficial to host immunity, had the largest inverse correlations with PACS at 6 months. PACS patients also had distinct gut metabolome compared to controls. Different patterns of gut microbiota were also seen in different PACS symptoms. Moreover, baseline *Blautia wexlerae* and *Bifidobacterium longum* had an inverse correlation with PACS at 6 months, whereas two *Actinomyces* species and *Atopobium parvulum* exhibited a positive correlation. Therefore, the dysbiosis of the gut microbiota may be involved in PACS development, though the exact extent of this involvement remains to be further investigated. 

### 2.3. Potential Mechanisms Underlying the Gut Microbiota and SARS-CoV-2 Infection Outcomes

The gut microbiota were linked to SARS-CoV-2 infection in several ways (Figure 1). Firstly, animal studies have shown that the gut microbiota had a dynamic relationship with the angiotensin-converting enzyme [2] (ACE2), which serves as the cell entry receptor for SARS-CoV-2 [39] and is abundantly expressed in enterocytes along the small intestine [40]. On one hand, ACE2 could influence gut microbiota ecology through the deregulation of ACE2-regulated uptake of tryptophan in the small intestine, which in turn affects downstream manifestations [41]. On the other hand, gut microbiota could modulate ACE2 expression in the gut [42]. One study found that *Coprobacillus* was associated with the up-regulation of ACE2 while some *Bacteroides* bacterium and species such as *Bifidobacterium longum* were associated with the downregulation of ACE2 [43]. In particular, four of the *Bacteroides* species found in this study were shown to be negatively correlated with fecal SARS-CoV-2 load in human subjects [13]. This suggested that the gut microbiota can potentially influence the disease course and severity of COVID-19 by mediating ACE2-dependent SARS-CoV-2 entry. Further studies on human subjects, however, are needed to determine the exact mechanisms. 

In addition, metabolites produced by the gut microbiota are also likely to play important roles in the immune response against SARS-CoV-2 infection. Examples of such metabolites include tryptophan, SCFAs, and secondary bile acids. Tryptophan and the metabolites derived from it (such as kynurenine and indoles) are important in changing the functions of various immune cells (such as regulatory T cells) and mediating inflammation [44]. A decrease in tryptophan was found to be associated with more severe COVID-19 symptoms [45,46]. This suggests that tryptophan metabolism may be another one of the links between the gut microbiota and COVID-19 development. 

SCFAs produced by the gut microbiota, such as acetate, propionate, and butyrate, can enter systemic circulation through passive diffusion or active transport by the gut epithelial cells, where they then modulate local and systemic inflammation and immune responses [47]. In particular, butyrate has been observed to support the integrity of the gut barrier and gut homeostasis and can exert anti-inflammatory effects through the inhibition of histone deacetylase in various immune cells and the activation of nuclear factor-κB (NF-κB), which in turn reduces the production of proinflammatory cytokines [48]. As mentioned previously, butyrate-producing species, especially *F. prausnitzii*, have been frequently observed to be depleted in COVID-19 patients, especially in those with more severe and persistent symptoms [6,11,13,14,25,38]. Indeed, *F. prausnitzii* has also been observed to be reduced in other inflammatory diseases, such as Crohn’s disease [49]. These observations support the notion that alterations in SCFA production, particularly butyrate, due to gut microbiota dysbiosis may exacerbate inflammation and thus the severity of COVID-19. Supplementation of the gut microbe species involved and SCFAs themselves can be a potential therapeutic strategy as an adjuvant to reduce the symptoms brought by COVID-19. In fact, this strategy has been tested on animal models for the treatment of inflammatory bowel diseases, such as colitis in rat models, with promising results [50,51].

Similarly, secondary bile acids have also been shown to be able to inhibit NF-κB signaling pathways, inhibit IL-17 expressing helper T cells, and enhance differentiation of regulatory T cells [52]. In the context of COVID-19, secondary bile acids were found to be significantly associated with the progression of respiratory failure and patient’s survival [53]. Ursodeoxycholic acid has also been suggested as a therapeutic agent for the prevention of cytokine storms in COVID-19 management by inhibiting the production of pro-inflammatory cytokines [54,55]. In line with this, one study found that the abundance of *Collinsella*, a major producer of ursodeoxycholate, was inversely correlated with COVID-19 mortality [56]. *Collinsella* had also been found to be significantly depleted in patients with PACS [38]. Therefore, secondary bile acids produced by the gut microbiota may very likely be involved in modulating the presentation of COVID-19 in different individuals. These metabolites were also important in mediating the crosstalk between the gut and other organs. One such crosstalk that is important in COVID-19 is that between the gut and the lungs, also known as the “gut-lung axis”. Through this axis, intact or fragmented gut bacteria, as well as their metabolites such as SCFAs, can cross the intestinal barrier and modulate the local immune response in the lungs via systemic circulation [57]. Studies have also shown that the gut microbiota could influence the expression of type I interferon receptors in respiratory epithelial cells, thereby mediating the secretion of IFNɑ and IFNβ and restricting viral replication [58]. Through the gut–lung axis, gut microbiota dysbiosis may potentially influence respiratory symptoms in COVID-19. In addition, the crosstalk between the gut and the brain, termed the “gut-brain axis”, may have been involved in the development of neuropsychiatric symptoms in COVID-19. Dysbiosis of the gut microbiota has been observed in various neuropsychiatric disorders that may be present in COVID-19 patients, such as anxiety, depression, and dementia [47]. SCFAs produced by the gut microbiota can bind to G protein-coupled receptors in the brain to modulate neuronal activity and mediate brain immunity [47,59]. Therefore, SCFA deficiency due to dysbiosis could have contributed to inflammation in the brain and other neuropsychiatric complications seen in COVID-19. 

## 3. Gut Microbiota and Vaccine Immunogenicity

The major mechanism of vaccination in protecting against infectious pathogens is via the stimulation of B cells to produce antigen-specific antibodies and inducing immune memory, though cell-mediated immunity provided by T-cells is also important in some cases [60]. Yet, it has been observed that the B cell and T cell responses to vaccination can be highly variable among different individuals, the reasons of which are still not yet fully understood. Intrinsic host factors including age, sex, genetics, and comorbidities contribute to variations in immune responses to vaccination [61]. Vaccine immunogenicity is notably lower in infants [62,63] and in the elderly [64] due to their weaker immune systems. The use of immunosuppressive drugs and therapies may also hinder the immune response to vaccinations, such as certain disease-modifying antirheumatic drugs (DMARDS) [65] and monoclonal antibody therapies used in inflammatory bowel diseases [66].

Recently, gut microbiota have been postulated to play a key role in vaccine immunogenicity. Several possible pathways between the gut microbiota and vaccine immunogenicity have been proposed (Figure 1). One such pathway is via the activation of pattern recognition receptors (PRRs), such as Toll-like receptors (TLRs) and NOD-like receptors (NLRs), which regulate the activities of antigen-presenting cells (APCs) [67]. One study investigated the causal link between TLR5 and trivalent inactivated influenza vaccine (TIV)-induced humoral immune response in mice found that the microbiota had significant impacts on plasma B cell response to vaccination through TLR-5 sensing of bacterial flagellin [68]. Specifically, the co-injection of flagellin and TIV into antibiotic-treated mice was able to rescue post-TIV antibody response to that seen in untreated mice. This was because flagellin enhanced the presence of short-lived plasma cells, which was important for early antibody responses to vaccinations. However, this effect of TLR-5-mediated sensing of microbiota on antibody response was not seen in adjuvanted vaccines or live-attenuated yellow fever vaccines. Another study showed that Nod2-mediated recognition of bacterial peptidoglycan molecules was important in the mucosal adjuvant activity of the cholera toxin [69]. Lipopolysaccharides (LPSs) secreted by the gut microbiota could be sensed by TLR-4, whose activation promotes antibody production and type 1 T helper cells (Th1), thereby producing an adjuvant effect to vaccination [67]. Apart from PRR-recognized molecules, the gut microbiota are also capable of producing metabolites that can potentially modulate immune responses. Examples include SCFAs and secondary bile acids, the former of which have been shown to be able to increase oxidative phosphorylation, glycolysis, and fatty acid synthesis for energy production in B cells [70], while the latter was suggested to be negatively correlated with inflammatory signatures in the influenza vaccine study previously described [71].

### 3.1. Gut Microbiota and Non-COVID-19 Vaccine Immunogenicity

The association between the composition of infant fecal microbiota and some orally administered vaccinations, in particular oral rotavirus (ORV), has been investigated in a number of studies (Table 2). Several studies on ORV conducted in less-developed countries (LDCs) found that *Bacilli* and *Firmicutes* were positively correlated with ORV response, while *Bacteroides* and *Prevotella* were negatively correlated [72,73]. In another study, microbiota diversity was negatively correlated with ORV seroconversion in LDCs [74]. On the other hand, other studies have reported no significant association between the infant gut microbiome and ORV immunogenicity [75,76,77]. Conclusive evidence establishing the correlation and causality between the infant gut microbiome and ORV immunogenicity remains to be obtained. In adults, a randomized controlled trial (RCT) found a higher abundance of *Enterobacteriaceae* (*Proteobacteria*) and a lower abundance of *Bacteroidetes* at the time of vaccination was correlated with ORV boosting and fecal shedding of rotavirus [78]. Nevertheless, evidence on the association of the gut microbiome to other orally administered vaccines, including oral polio vaccines (OPVs) [79,80] and oral cholera vaccines (OCVs) [81,82] were also often contradictory. At the phylum level, *Proteobacteria* were observed to be enriched in rotavirus vaccine responders in two studies, while at the species level, *Escherichia coli* was reported to be enriched in two studies (rotavirus and cholera vaccines) (Table A2). On the other hand, *Bacteroides* were noted to have a lower relative abundance in vaccine responders in two studies (cholera and rotavirus vaccines) (Table A2).

Several studies have attempted to observe changes in influenza vaccine immunogenicity following modulation of the gut microbiota through different means, such as the use of antibiotics, probiotics, and synbiotics. One notable study assessed the effect of broad-spectrum antibiotics on immune response to influenza vaccine in adults [71]. It was found that at early time points, *Enterobacteriaceae* and *Streptococcaceae* were more abundant, while *Lachnospiraceae*, *Ruminococcaceae*, *Bacteroidaceae*, and *Veillonellaceae* were diminished in antibiotic treatment group. Alpha diversity and beta diversity were heavily compromised after antibiotic use with only partial recovery at six months. Antibiotic-treated subjects had significantly reduced concentrations of vaccine-induced H1N1 strain-specific antibodies, increases in inflammatory signaling and disturbed plasma metabolome. In particular, there was a 1000-fold reduction in the plasma level of lithocholic acid, which was strongly correlated with inflammatory blood transcriptional modules, implying a role of secondary bile acids in modulating inflammation. Another study which used probiotics found that in the probiotic group there was enrichment of bacteria capable of short chain fatty acids (SCFAs) production and also linked to anti-inflammatory effects [86]. On the other hand, two studies which used synbiotics did not find an association between the use of synbiotics and vaccine-induced response by B and T cells [87] as well as natural killer cells [88]. These negative findings were likely related to immunosenescence among the elderly.

### 3.2. Gut Microbiota and COVID-19 Vaccine Immunogenicity

The factors affecting COVID-19 vaccine immunogenicity are found to be similar to those in other vaccines. Studies have shown that demographic factors including male gender [89,90,91,92,93], older age [89,90,91,92,93,94], immunosuppressive conditions, therapies [89,91] including monoclonal antibody therapies [66,89], DMARDs [65,89], hematological cancer [89], long courses of steroids [89], and comorbidities such as obesity [89,95], diabetes [89,91,94], hypertension [89,91,93,94], heart diseases [89,91], and liver diseases including non-alcoholic fatty liver disease [96,97] were inversely associated with post-vaccination antibody levels.

Intuitively, gut microbiota which are one of the factors affecting vaccine immunogenicity may also have influences on COVID-19 vaccine immunogenicity. Emerging evidence has shown that the gut microbiota may be associated with COVID-19 vaccine immunogenicity (Table 2). One prospective study investigated the associations between recent use of antibiotics and immunogenicity within 6 months of receiving the BNT162b2 vaccine in Hong Kong [98]. Antibiotic use was defined as the use of any antibiotics within six months before vaccination. A total of 312 BNT162b2 recipients were included, with 29 of them being antibiotic users according to the definition. It was observed that there was a trend toward antibiotic users having lower seroconversion rate and median antibody level than antibiotic non-users, and this trend was diminished following two doses of BNT162b2 compared to that after one dose. Multivariate regression found that recent antibiotic use was significantly associated with 74% lower chance of seroconversion after one dose of BNT162b2 vaccination, but not after two doses. Although this study did not collect any stool samples for analysis of fecal microbiota, it was likely that there was underlying gut microbiota dysbiosis which could have influenced COVID-19 vaccine immunogenicity in this circumstance. However, this study was limited by a small sample size of the antibiotics cohort, which made stratified analysis not possible. Data at further timepoints as well as on other SARS-CoV-2 variants were also not collected.

Another prospective study conducted on 43 infliximab-treated inflammatory bowel disease (IBD) patients also associated gut microbiota perturbations with serological response to two COVID-19 vaccinations, which were BNT162b2 or ChAdOx1 [83]. It was found that the gut microbiota of recipients with below-average antibody concentrations had lower beta diversity compared to those with above-average antibody concentrations. Of the bacteria found to be differentially abundant between above and below average responders, *Bilophila* was associated with above-average response, while *Streptococcus* was associated with below-average response. This study also profiled the fecal metabolome of the patients, and it was found that above average responders had significantly higher levels of trimethylamine, omega-muricholic acid, and ursodeoxycholic acid, whereas below-average responders were significantly more enriched in succinate, phenylalanine, phenylacetate, as well as the bile acids taurolithocholate and taurodeoxycholate. Subsequent correlation analysis associated *Streptococcus* with phenylalanine, both of which were inversely correlated with antibody response. On the other hand, acetate, which is a SCFA, was negatively correlated with *Streptococcus*. *Bilophila* was associated with trimethylamine, and both of them were positively correlated with antibody response. *Bilophila* was additionally associated with methylamine and chenodeoxycholic acid, and negatively associated with the SCFA valerate. Several amino acids were also found to be correlated with *Dialister.* Although this study was limited by a relatively small cohort focused on immunosuppressed IBD patients and lack of analysis of factors that could affect the gut microbiota and metabolome, it nonetheless showed the potential link of certain gut microbial species and metabolites, such as trimethylamine, SCFAs, and bile acids with COVID-19 vaccine immunogenicity. 

A prospective study investigated the associations between the gut microbiota and COVID-19 vaccine immunogenicity on 138 participants in Hong Kong, 108 of which received BNT162b2 vaccines, while the remaining received CoronaVac inactivated virus vaccine [84]. The antibody levels were measured at baseline and one month after the second dose of vaccination. This study found that for both vaccine types, the gut microbiome after the second dose of vaccination had significantly lower alpha diversity and experienced shifts in beta diversity compared to baseline. Participants of each vaccine type were divided into high-responders (with SARS-CoV-2 surrogate virus neutralization test (sVNT) inhibition higher than 60%) and low-responders for the remaining. Baseline gut microbiome was able to predict the immune response at one-month post-vaccination. In the baseline microbiome of the CoronaVac group, high responders had more abundant *Bifidobacterium adolescentis*, which was significantly correlated with sVNT% and the abundance of carbohydrate metabolism pathways. In contrast, low-responders were more enriched with *Bacteroides vulgatus*, *Bacteroides thetaiotaomicron*, and *Ruminococcus gnavus*, the former two of which were also positively correlated with the abundance of L-ornithine biosynthesis II pathway. In addition, *B. adolescentis* remained persistently high while *B. vulgatus* remained persistently low from baseline to one month after second dose in high responders. On the other hand, in the baseline gut microbiome of the BNT162b2 group, *B. adolescentis* was again persistently low among low responders. In addition, the highest-tier responders were significantly enriched with *Roseburia faecis*, *Eubacterium rectale*, and two *Bacteroides* species. In particular, *R faecis* expressed flagella and fimbriae, which were positively associated with antibody response. This study also found that BMI was correlated with sVNT levels. Overweight or obesity could attenuate the beneficial effect on immune response by four bacterial species, namely *B. adolescentis*, *Butyricimonas virosa*, *Adlercreutzia equolifaciens*, and *Asaccharobacter celatus*. Interestingly, the gut microbiome was also associated with vaccine-related adverse events. Those who reported adverse effects after BNT162b2 vaccination were found to have significantly decreased species richness. Enrichment of *Prevotella copri, Megamonas funiformis*, and *Megamonas hypermegale* were associated with fewer adverse effects for both vaccine types, suggesting that they may have anti-inflammatory effects. In summary, this study demonstrated that the gut microbiota, especially species involved with immune modulations, were associated with COVID-19 vaccine immunogenicity and adverse effects.

Another prospective study recruiting 207 subjects conducted in China investigated the correlation between the gut microbiome and metabolome with immune response to BBIBP-CorV-inactivated COVID-19 vaccine [85]. The results showed that the gut microbiome at baseline and after vaccination was different. Baseline gut microbiome was significantly associated with measures of immunity, including monocytes, total T cells, cytotoxic T cells, helper T cells, IL-8 concentration, and TNF-α. Significant changes in gut microbial functional profile were also observed. Importantly, pathways related to fatty acid biosynthesis and fermentation to SCFAs were found to be decreased after the second dose of vaccination compared to baseline. These pathways at baseline were significantly associated with lymphocytes, T cell ratio, and IL-8 concentration after the second dose. In particular, pathways involved in the fermentation to SCFAs at baseline were positively correlated with antibody response after second dose, and these pathways were mainly contributed to by *Anaerostipes hadrus*. The participants were then divided into high and low groups based on their ACE2-RBD inhibiting antibody concentrations at day 42. *Collinsella aerofaciens*, *Fusicatenibacter saccharivorans*, *Eubacterium ramulus*, and *Veillonella dispar* were significantly more abundant in the high group, while in the low group *Lawsonibacter asaccharolyticus* was enriched. Fecal and serum levels of the SCFAs acetic acid and butyric acid at day 42 were also positively correlated with antibody response at day 42. This study was limited by a lack of an independent validation cohort, long-term follow up to investigate the impact of the gut microbiome and metabolome on antibody decline, and analysis of factors that can influence gut microbiome and metabolome, such as diet. Nonetheless, it once again demonstrated that the gut microbiota and their metabolites, especially SCFAs, could be associated with COVID-19 vaccine immunogenicity. 

Notably, *Eubacterium* was the only genus reported to be enriched in more than one study (including *E. rectale* and *E. ramulus* at the species level) (Table A2). Otherwise, the above studies have identified different gut bacterial genera and species that were associated with vaccine response toward COVID-19 vaccine. This was possibly due to different populations being included. Apart from the genus *Bifidobacterium*, which was reported to be depleted in responders of both COVID-19 vaccines and non-COVID-19 vaccines, there were also no other bacterial genus or species in common between these two populations (Table A2).

There are several common limitations of the above studies. First, the study populations were mainly focused on the Chinese population and may not be representative of other ethnicities due to inherent geographical variations. It is possible that studies conducted in other ethnicities and geographical areas may find other species that are potentially involved in COVID-19 vaccine immunogenicity. The gut microbiota are also influenced by diets and lifestyle habits, which in turn contribute to variations across different populations from different geographical areas, which is particularly substantial between high-income countries (HICs) and low- and middle-income countries (LMICs) with large differences in socioeconomic status measured in degrees of urbanization and industrialization [7]. Such differences may be part of the reason why the above studies did not share any particular bacterial species in common that may be associated with COVID-19 vaccine immunogenicity, as they were each carried out in different geographical areas. Second, the time period of most studies may not have been sufficient to investigate the potential associations between the gut microbiota and the long-term antibody levels following two doses of vaccination, which may be important in light of the waning of antibody levels over time. Moreover, all of the studies only included up to the second dose of vaccination. Further studies are needed to determine whether the changes in gut microbiota composition and metabolites observed are also similar following booster doses. It may also be worthwhile to carry out studies targeting vulnerable populations, such as young children and the elderly as they have been shown to have weaker immune responses to vaccinations and possibly different gut microbiota composition to adults. Third, these studies were all carried out on COVID-19-negative participants. As the gut microbiota and their metabolites were separately shown to be associated with SARS-CoV-2 infection outcomes and severity, it would be interesting to see whether the gut microbiota could have also modulated COVID-19 severity and outcomes post-vaccination, which was another important indicator of the protective effect provided by COVID-19 vaccines. Lastly, due to the rapid emergence of new variants of SARS-CoV2, particularly the Omicron variants with strong immune escape potentials, it remains to be determined whether these observations on gut microbiota are still valid.

All these can be valuable in providing information to guide the development of gut microbiota-based interventions, such as probiotics and synbiotics, to improve COVID-19 vaccine immunogenicity among the general population. The generated findings may also guide future studies in the further exploration of the role of gut microbiota in the immune response and adverse events seen in other vaccines.

## 4. Conclusions

With the gut microbiota gaining more research interest in recent years, it has emerged that the effects exerted by the gut microbiota could act far beyond the confines of the gastrointestinal tract. Emerging observations and evidence have shown that the gut microbiota could contribute to our body response to SARS-CoV-2 infections, and, through their immunomodulatory effects and interactions with different organs, mediate the disease manifestations and severity of COVID-19. The gut microbiome and metabolome are also crucial in influencing the immunogenicity for different vaccinations, including COVID-19 vaccines. In this regard, probiotic bacterial species such as *Bifidobacterium* and immunomodulatory metabolites such as SCFAs are particularly important. Although studies on this aspect are still limited in terms of number, sample sizes, short-term follow-up, and scope of outcomes, these preliminary findings have nonetheless provided novel discoveries which may pave the way for future breakthroughs in utilizing accessible and affordable interventions to the gut microbiome and metabolomics in order to improve the protective effects of vaccinations against SARS-CoV-2 and other infectious pathogens, and thus opening a new chapter in the prevention of infectious diseases on individual and population levels. 

## Figures and Tables

**Figure 1 microorganisms-11-00452-f001:**
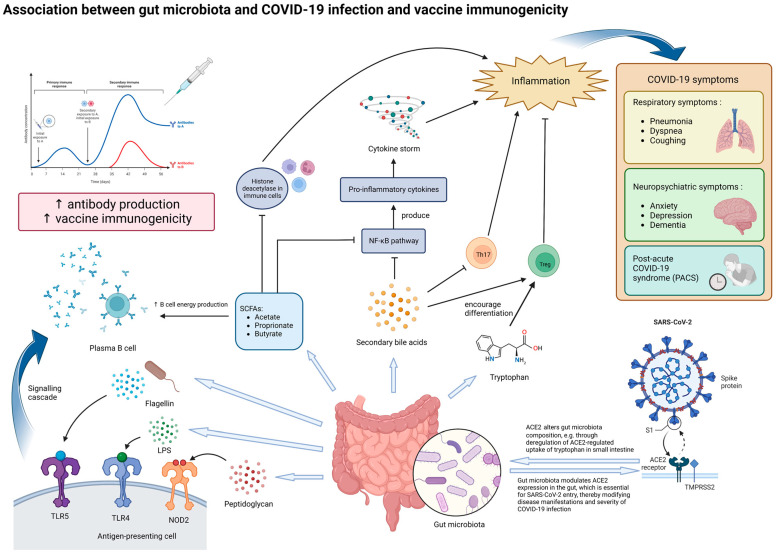
Potential mechanisms underlying the effect of gut microbiota on SARS-CoV-2 infection and vaccine immunogenicity. The gut microbiota and its metabolites, particularly those with immunomodulatory properties, can influence both the manifestations of COVID-19 and vaccine immunogenicity. In the context of COVID-19, dysbiosis of the gut microbiota may increase the severity of inflammation and various symptoms through modulating ACE2 expression in enterocytes and altered secretion of immunomodulatory molecules, such as tryptophan, SCFAs and secondary bile acids. Dysbiosis may potentially contribute to the production of cytokine storms, which produce more severe symptoms. In the long run, dysbiosis may be associated with persistent COVID-19 symptoms and inflammation, termed as post-acute COVID-19 syndrome (PACS). In the context of vaccine immunogenicity, lipopolysaccharides (LPSs), flagellin, peptidoglycan, and SCFAs secreted by the gut microbiota can enhance antibody production to vaccination by plasma B cells, thereby improving vaccine immunogenicity. Abbreviations: ACE2, angiotensin-converting enzyme 2; LPS, lipopolysaccharide; NOD2, nucleotide-binding oligomerization domain-containing protein 2; TLR-4, Toll-like receptor 4; TLR-5, Toll-like receptor 5; SCFA, short-chain fatty acid; NF-κB, nuclear factor-κB; Th17, T helper 17 cells; Treg, regulatory T cells.

**Table 1 microorganisms-11-00452-t001:** Summary of observational studies on gut microbiota alterations in COVID-19 patients.

Study	Study Participants	Sequencing Method	Gut Microbiota Alterations during SARS-CoV-2 Infection
**Adult Patients**			
Gaibani et al. [28]	69 COVID-19 patients and 69 healthy controls from Italy	16S rRNA sequencing	Reduced diversity in COVID-19 patients Enriched in COVID-19 patients: *Enterococcus*, *Staphylococcus*, *Serratia*, *Collinsella*, *Lactobacillus*, *Parabacteroides*, *Lactococcus*, *Phascolarctobacterium*, *Odoribacter*, *Actinomyces*, *Methanobrevibacter*, and *Akkermansia* Enriched in controls: *Bacteroidaceae* (i.e., *Prevotella* and *Bacteroides*), *Lachnospiraceae* (i.e., *Coprococcus*, *Blautia*, *Roseburia*, and *Lachnospira*), and *Ruminococcaceae* (i.e., *Faecalibacterium*, *Ruminococcus*, *Oscillospira*, and *Anaerofilum*) *Enterococcus* was particularly overrepresented in patients who developed bloodstream infections and admitted into ICU
Gu et al. [26]	30 COVID-19 patients, 24 H1N1 patients, and 30 healthy controls from China	16S rRNA sequencing	Reduced diversity in COVID-19 patients Enriched in COVID-19 patients: *Streptococcus*, *Rothia*, *Veillonella*, *Erysipelatoclostridium*, and *Actinomyces* Depleted in COVID-19 patients (compared to controls): *Ruminococcaceae* family and *Lachnospiraceae* family (*Fusicatenibacter*, *Anaerostipes*, *Agathobacter*, *unclassified Lachnospiraceae*, and *Eubacterium hallii* group) Enriched in healthy controls: *Romboutsia*, *Faecalibacterium*, *Fusicatenibacter*, and *E. hallii* group
Ren et al. [29]	36 COVID-19 patients and 70 healthy controls from China	16s rRNA sequencing	Reduced diversity in COVID-19 patients Enriched in COVID-19 patients: *Leptotrichia*, *Selenomonas*, *Megasphaera*, *Campylobacter*, and *Granulicatella* (*Leptotrichia* and *Selenomonas are lipopolysaccharide (LPS)-producing bacteria*) Depleted in COVID-19 patients (compared to controls): *Peptostreptococcus*, *Haemophilus*, *Fusobacterium*, *Streptococcus*, and *Porphyromonas* (*Porphyromonas* and *Fusobacterium* are butyrate-producing bacteria)
Xu et al. [30]	35 COVID-19 patients, 19 healthy controls, and 10 non-COVID patients with other diseases from China	16S rRNA sequencing	Reduced diversity in COVID-19 patients
Mizutani et al. [31]	22 COVID-19 patients and 40 healthy controls from Japan	16S rRNA sequencing	Enriched in COVID-19 patients: *Bifidobacterium*, *Bacteroides*, *Parabacteroides*, and *Escherichia-Shigella* Depleted in COVID-19 patients: *Faecalibacterium*, *Subdoligranulum*, *Dorea*, and *Enterobacter*
Rafiqul Islam et al. [32]	22 COVID-19 patients and 15 healthy controls from Bangladesh	16S rRNA sequencing	Enriched in COVID-19 patients: *Escherichia*, *Shigella*, *Enterococcus*, *Bacteroides*, and *Bifidobacterium*
Reinold et al. [14]	117 COVID-19 patients and 95 hospitalized patients as controls from Germany	16S rRNA sequencing	Enriched in COVID-19 patients: *Bacteroides*, *Enterobacteriaceae*, and *Campylobacteraceae* Enriched in non-COVID-19 controls: *Bifidobacterium*, *Collinsella*, *Streptococcus*, and *Corynebacterium* Negatively correlated with COVID-19 severity: *Faecalibacterium* and *Roseburia*
Tang et al. [33]	57 COVID-19 patients from China	qPCR	Enriched in COVID-19 patients: *Enterococcus*, *Enterobacteriaceae* Depleted in COVID-19 patients: Butyrate-producing bacteria including *Faecalibacterium prausnitzii*, *Clostridium butyricum*, *Clostridium leptum*, and *Eubacterium rectale*
Tao et al. [27]	62 COVID-19 patients, 33 seasonal flu patients, and 40 healthy controls from China	16s rRNA sequencing	Reduced diversity in COVID-19 patients Enriched in COVID-19 patients (compared to controls): *Streptococcus*, *Clostridium*, *Lactobacillus*, and *Bifidobacterium* Depleted in COVID-19 patients: *Bacteroidetes*, *Roseburia*, *Faecalibacterium*, *Coprococcus*, and *Parabacteroides*
Wu et al. [34]	53 COVID-19 patients and 76 healthy controls from China	16S rRNA sequencing	Reduced diversity in COVID-19 patients Enriched in COVID-19 patients: *Streptococcus*, *Weissella*, *Enterococcus*, *Rothia*, *Lactobacillus*, *Actinomyces*, *Granulicatella*, *Clostridium citroniae*, *Bifidobacterium longum*, and *Rothia mucilaginosa* Depleted in COVID-19 patients: *Blautia*, *Coprococcus*, *Collinsella*, *Bacteroides caccae*, *Bacteroides coprophilus*, *Blautia obeum*, and *Clostridium colinum*
Yeoh et al. [11]	100 COVID-19 patients and 78 non-COVID-19 controls from Hong Kong	Shotgun metagenomic sequencing	Enriched in COVID-19 patients: *Ruminococcus gnavus, Ruminococcus torques*, and *Bacteroides dorei* Depleted in COVID-19 patients: *Bifidobacterium adolescentis, Faecalibacterium prausnitzii*, and *Eubacterium rectale* Negatively correlated with COVID-19 severity: *F. prausnitzii* and *Bifidobacterium bifidum*
Nagata et al. [12]	112 COVID-19 patients and 112 non-COVID-19 controls from Japan	Shotgun metagenomic sequencing	Enriched in COVID-19 patients: *Ruminococcus torques* Depleted in COVID-19 patients: SCFA producers including *Bifidobacterium*, *Dorea*, *Roseburia*, and *Butyricicoccus* species
Li et al. [35]	47 COVID-19 patients and 19 controls from China	Shotgun metagenomic sequencing	Enriched in COVID-19 patients: *Bacteroides stercoris*, *Bacteroides vulgatus*, *Bacteroides massiliensis*, *Bifidobacterium longum*, *Streptococcus thermophilus*, *Lachnospiraceae bacterium 5163FAA*, *Prevotella bivia*, *Erysipelotrichaceae bacterium 6145*, and *Erysipelotrichaceae bacterium 2244A* Depleted in COVID-19 patients: *Clostridium nexile*, *Streptococcus salivarius*, *Coprococcus catus*, *Eubacterium hallii*, *Enterobacter aerogenes*, and *Adlercreutzia equolifaciens* Negatively correlated with COVID-19 severity: *Roseburia inulinivorans*, *Bacteroides faecis*, *Bifidobacterium bifidum*, *Parabacteroides goldsteinii*, *Lachnospiraceae bacterium 9143BFAA*, and *Megasphaera* sp. Positively correlated with COVID-19 severity: *Paraprevotella* sp., *Streptococcus thermophilus*, *Clostridium ramosum*, and *Bifidobacterium animalis*
**Pediatric Patients**			
Romani et al. [22]	68 COVID-19 patients, 4 patients with multisystem inflammatory syndrome in children (MIS-C), 16 non-COVID-19 controls from Italy	16S rRNA sequencing	Enriched in COVID-19 patients: *Faecalibacterium*, *Fusobacterium*, and *Neisseria* Depleted in COVID-19 patients: *Bifidobacterium*, *Blautia*, *Ruminococcus*, *Collinsella*, *Coprococcus*, *Eggerthella*, and *Akkermansia* Enriched in MIS-C patients: *Veillonella*, *Clostridium*, *Dialister*, *Ruminococcus*, and *Streptococcus* Depleted in MIS-C patients: Bifidobacterium, Blautia, Granulicatella, and Prevotella
Xu et al. [23]	9 COVID-19 patients and 14 healthy controls from China	16S rRNA sequencing	Enriched in COVID-19 patients: *Pseudomonas, Herbaspirillum*, and *Burkholderia*
Nashed et al. [24]	13 children from USA with SARS-CoV-2 positive fecal samples	16S rRNA sequencing	Depleted in positive samples: *Bifidobacterium bifidum* and *Akkermansia muciniphila*
Suskun et al. [25]	64 COVID-19 patients, 25 MIS-C patients, and 19 healthy controls from Turkey	16S rRNA sequencing	Enriched in MIS-C patients: *Bacteroides uniformis*, *Bacteroides plebeius*, *Clostridium ramosum*, *Eubacterium dolichum*, *Eggerthella lenta*, *Bacillus thermoamylovorans*, *Prevotella tannerae*, and *Bacteroides coprophilus* Depleted in MIS-C patients: *Faecalibacterium prausnitzii*

Abbreviations: ICU, intensive care unit; qPCR, quantitative polymerase chain reaction; SCFA, short-chain fatty acid; MIS-C, multisystem inflammatory syndrome in children.

**Table 2 microorganisms-11-00452-t002:** Summary of clinical studies investigating association between gut microbiota and vaccine immunogenicity (non-COVID-19 and COVID-19 vaccines).

Study	Study Type and Sequencing Method	Primary Outcome	Major Findings	Limitations
**Non-COVID-19 Vaccines**				
Harris et al. [72]	Nested, case–control study on 78 Ghanian infants (39 oral rotavirus vaccine (ORV) responders and 39 ORV non-responders) with comparison to 154 Dutch infants assumed to be ORV responders Sequencing method: HITChip microarray	Whether intestinal microbiome composition in infants correlated with ORV efficacy, and whether the intestinal microbiota composition was different in ORV responders and non-responders	** *Gut microbiota:* ** Bacilli phylum, in particular bacteria related to *Streptococcus bovis*, was significantly correlated to high ORV response Bacteroidetes phylum, specifically bacteria related to *Bacteroides* and *Prevotella species*, was significantly correlated to a lack of ORV response Dutch infants’ overall fecal microbiota composition was significantly more similar to that of Ghanian ORV responders than that of non-responders	No actual ORV immunogenicity data for Dutch infant cohort No specific data on potential confounders, such as breastfeeding and delivery practice, and levels of maternally derived rotavirus antibodies in Ghanian infants Anti-RV IgA response may not be sufficient surrogate for vaccine efficacy. Larger sample sizes and follow-up are required Only intestinal bacterial populations were analyzed Correlative associations between microbiome and ORV response only, not causative
Harris et al. [73]	Nested, matched case–control study between 10 Pakistini ORV responders, 10 Pakistini ORV non-responders, and 10 healthy Dutch infants assumed to be ORV responders Sequencing method: HITChip microarray	Whether intestinal microbiome composition in infants correlated with ORV efficacy, and whether the intestinal microbiota composition was different in ORV responders and non-responders	** *Gut microbiota:* ** Firmicutes, in particular bacteria belonging to *Clostridium* cluster XI and Proteobacteria, were significantly enriched in Pakistini ORV responders. Enrichment of *Proteobacteria* was also observed in matched Dutch infants Gram-negative bacteria related to *Serratia* and *Escherichia coli* were positively associated with vaccine response	Small sample size Does not account for variation of infant gut microbiome over time
Parker et al. [74]	Prospective multicenter cohort study on infants receiving ORV in India (*n* = 307), Malawi (*n* = 119), and the UK (*n* = 60) Sequencing method: 16S rRNA sequencing	Effect of maternal antibodies, environmental enteric dysfunction (EED) markers and bacterial gut microbiota development on RRV response among infants from India, Malawi, and the UK	** *Gut microbiota:* ** Increased microbiota diversity is negatively correlated with ORV response in infants from Indian and Malawi, but not the UK	RV-IgA was suboptimal correlate of vaccine protection Smaller-than-target sample size in Malawi cohort
Robertson et al. [75]	Prospective cohort study on 158 infants from rural Zimbabwe previously enrolled in the Sanitation Hygiene Infant Nutrition Efficacy (SHINE) trial Sequencing method: Whole metagenome shotgun sequencing	Whether alterations in the composition of the fecal microbiome are associated with ORV immunogenicity	** *Gut microbiota:* ** No significant differences in species composition, alpha, and beta diversity by ORV seroconversion status	Small sample size Seroconversion and seropositivity may not be accurate correlates of vaccine protection No control over the administration, dosing, or timing of vaccinations; microbiome assessment was therefore independent of vaccine administration
Fix et al. [76]	Prospective study on 50 children receiving ORV from Nicaragua Sequencing method: 16S rRNA sequencing	Evaluate the relationship between gut microbiome community structure and response to ORV	** *Gut microbiota:* ** No significant difference in gut microbiome composition between ORV responders and non-responders	Small sample size Lack of longitudinal assessment of variation of gut microbiome composition over a period of time
Parker et al. [77]	Nested case–control study on 170 infants receiving ORV from India Sequencing method: 16S rRNA sequencing	Whether failure of seroconversion after ORV vaccination is associated with elevated pathogen burden and altered bacterial microbiota composition	** *Gut microbiota:* ** No significant difference in gut microbiome composition between ORV responders and non-responders	Did not account for potential confounders that may influence microbiota composition, such as mode of delivery and antibiotic exposure
Harris et al. [78]	Randomized-controlled, open label trial on 63 adults (21 for each group: control, narrow-spectrum antibiotics, broad-spectrum antibiotics) receiving ORV Sequencing method: 16S rRNA sequencing	Whether modulation of adult gut microbiome can affect ORV immunogenicity	** *Gut microbiota:* ** Higher abundance of Enterobacteriaceae (Proteobacteria) and a lower abundance of Bacteroidetes at the time of vaccination was correlated with ORV boosting	Adult gut microbiome was largely different from that of infants, which was often more affected by rotavirus infection Antibiotics may have altered immunity through off-target effects
Zhao et al. [79]	Randomized, double-blind trial on 107 infants from China receiving different sequential immunization schedules combining inactivated polio vaccine (IPV) and oral polio vaccine (OPV) Sequencing method: 16S rRNA sequencing	Relationship between composition of intestinal microbiota and gut mucosal IgA response to polio vaccine	** *Gut microbiota:* ** Higher abundance of *Firmicutes* and lower abundance of *Actinobacteria* were observed in IgA-negative infants Higher gut microbiota diversity in IgA-negative infants at time of OPV inoculation	Modest sample size Did not account for other factors that may be attributed to mucosal antibody responses, such as genetic factors Effects of a single serotype of polio vaccine cannot be analyzed clearly because the stool samples were pooled across different schedules and IgA serotypes
Praharaj et al. [80]	Randomized, placebo-controlled trial on 120 infants receiving OPV from India (60 in each group: placebo or oral azithromycin) Sequencing method: 16S rRNA sequencing	Whether OPV response was associated with specific enterovirus serotypes or species, short-term changes in enteric virus burden or bacterial microbiota composition	** *Gut microbiota:* ** No significant difference in abundance of specific bacterial taxa according to OPV response	Biases in amplification efficiency may undermined the characterization of bacterial microbiota Effects of enteric viruses at low abundance may not have been characterized
Yuki et al. [81]	Phase 1 randomized clinical trial on 60 adult men from Japan receiving MucoRice-CTB oral cholera vaccine (OCV) Sequencing method: Metagenomics	To assess the safety, tolerability, and immunogenicity of MucoRice-CTB vaccine and the effect the gut microbiota have on immune response to the vaccine	** *Gut microbiota:* ** *Bacteroides* was significantly less abundant in responders while *E. coli* and *Shigella* was significantly more enriched Higher gut microbiota diversity in responders than non-responders	Outbreaks of enterotoxigenic *E. coli* in Japan may have influenced the outcomes of gut microbiota analysis
Chac et al. [82]	Randomized trial on 69 adults from Bangladesh receiving OCV Sequencing method: 16s rRNA sequencing	To investigate the relationship between the gut microbiota and responses to OCV	** *Gut microbiota:* ** Gut microbial diversity at the time of vaccination was not associated with memory B cell (MBC) responses to OCV Individuals with higher *Clostridiales* abundance and lower *Enterobacterales* abundance were more likely to develop MBC response	Small sample size with primarily female population More frequent fecal sampling is required for higher resolution on fluctuations of microbiota diversity over time after vaccination
**COVID-19 vaccine**				
Alexander et al. [83]	Prospective cohort study on 43 infliximab-treated inflammatory bowel disease (IBD) patients receiving either BNT162b2 or ChAdOx1 COVID-19 vaccine Sequencing method: 16S rRNA sequencing	Potential influences of gut microbiota composition and function on immune response to SARS-CoV-2 vaccination in immunosuppressed patients with IBD	** *Gut microbiota:* ** Beta diversity of gut microbiota was lower in recipients with below-average antibody concentrations *Bilophila* was associated with above average antibody response *Streptococcus* was associated with below average response ** *Gut metabolome:* ** Above average responders had significantly higher levels of trimethylamine, omega-muricholic acid, and ursodeoxycholic acid. In particular, trimethylamine was positively correlated with *Bilophila* Below average responders were significantly more enriched in succinate, phenylalanine, phenylacetate as well as the bile acids taurolithocholate and taurodeoxycholate. In particular, phenylalanine was correlated with *Streptococcus*	Small cohort focused on immunosuppressed IBD patients Lack of analysis of factors that could affect gut microbiota and metabolome
Ng et al. [84]	Prospective cohort study on 138 Hong Kong participants receiving either BNT162b2 or CoronaVac inactivated virus COVID-10 vaccine Sequencing method: Shotgun metagenomic sequencing	Potential associations of gut microbiota composition with immune responses and adverse effects in adults receiving COVID-19 vaccination	** *Gut microbiota:* ** For both vaccine types, significantly lower alpha diversity and also shifts in beta diversity were observed in gut microbiome following second dose In both vaccine types, high responders (sVNT > 60%) were persistently enriched in *Bifidobacterium adolescentis* from baseline to one month after second dose In baseline microbiome of CoronaVac group, high responders were enriched with *B. adolescentis*, while low responders were enriched with *Bacteroides vulgatus*, *Bacteroides thetaiotaomicron*, and *Ruminococcus gnavus* In baseline microbiome of BNT162b2 group, high-responders were enriched with *Eubacterium rectale*, *Roseburia faecis*, *B. thetaiotaomicron*, and *Bacteroides sp OM05-12* Beneficial effect on immune response by four bacterial species could be attenuated by obesity In both vaccine types, the abundances of *Prevotella copri*, *Megamonas funiformis*, and *Megamonas hypermegale* were inversely correlated with adverse events, suggesting possible anti-inflammatory effects * **Functional pathways:** * High responders had higher abundances of pathways related to carbohydrate metabolism, which was correlated with *B. adolescentis* Low responders had higher abundance of L-ornithine biosynthesis II pathway, which was correlated with *B. vulgatus* and *B. thetaiotaomicron*	Lack of long-term follow up to investigate the impact of gut microbiome and metabolome on antibody waning
Tang et al. [85]	Prospective cohort study on 207 Chinese participants receiving BBIBP-CorV inactivated COVID-19 vaccine Sequencing method: Metagenomic sequencing	Possible correlations between gut microbiota and metabolic functions with immune response to BBIBP-CorV inactivated COVID-19 vaccine	* **Gut microbiota:** * Baseline gut microbiome was significantly associated with measures of immunity High responders were enriched in *Collinsella aerofaciens*, *Fusicatenibacter saccharivorans*, *Eubacterium ramulus*, and *Veillonella dispar* Low responders were enriched in *Lawsonibacter asaccharolyticus* ** *Metabolome/functional pathways:* ** Fecal and serum levels of SCFAs were positively correlated with antibody response at day 42 Pathways related to fatty acid biosynthesis and fermentation to SCFAs at baseline were significantly associated with lymphocytes, T cell ratio, and IL-8 concentration after second dose Pathways involved in fermentation to SCFAs, which were mainly contributed to by *Anaerostipes hadrus*, were additionally associated with increased antibody response after second dose	Lack of independent validation cohort Lack of long-term follow up to investigate the impact of gut microbiome and metabolome on antibody waning Lack of analysis of factors that could affect gut microbiota and metabolome

## Data Availability

No new data were created or analyzed in this study. Data sharing is not applicable to this article.

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
