# Peer review of "Association between Gut Microbiota and SARS-CoV-2 Infection and Vaccine Immunogenicity"

_microorganisms, 2023, doi:10.3390/microorganisms11020452_

Round 1

Reviewer 1 Report

The literature on Covid-19 is of course vast, and reviews on certain aspects on this infection are welcome for those who want to be informed about where to delve deeper into the issues. This review is basically good, but could be slightly improved by adding a little bit of more summaries especially concerning the tables.

Table 1 is a good overview of studies relating microbiome composition to covid infection. It could be complemented by adding a summary table showing which of the results that are consistent, showing the same bacterial types being more or less abundant, and clearly pointing out discordant results.

Table 2 is more diverse and would benefit from more focus. There are a number of themes involved, and it becomes hard to see specific patterns. There are only two papers relating to antimicrobial use, one in the non-covid and one in the covid vaccine part. To be appropriate for a review more studies should be included, but since the coverage is already vast, I would suggest removing these two examples. Also, the non-covid vaccine part is diverse with different vaccines as examples, and divisions into subsets means also here few studies for each vaccine exemplified. To be able to make a useful summary, first different non-covid vaccines should be compared and common patterns noted, then these combined results should be compared to covid vaccine studies to provide a proper comparison. 

Figure 1 is nice, but information on who is behind the design and copyright information should be provided.

Reviewer 2 Report

In this original manuscript, the authors extensively discussed the role and observed shifts of the gut microbiota of COVID-19 and vaccine immunogenicity. It is known that o  gut microbiota and its metabolites could produce immunomodulatory effects on vaccine immunogenicity in the case of influenza and oral rotavirus vaccines. In this vein, based on an extended bibliography discuss the role of gut microbiota in influencing the immune responses to COVID-19 vaccinations. Yet, they discuss the possible involved mechanisms, such as the effects of lipopolysaccharides, flagellin, peptidoglycan , and short-chain fatty acids. All information on the potential mechanisms of the effect of gut microbiota on COVID-19 infection and vaccine immunogenicity is represented in an excellent figure which collects information. In the context of COVID-19, dysbiosis of the gut microbiota may increase the severity of inflammation and various symptoms through modulating ACE2 expression in enterocytes and altered secretion of immunomodulatory molecules, such as tryptophan, SCFAs, and secondary bile acids. They stated that intestinal microbiota mediates the disease severity of COVID-19 and impacts the immunogenicity for vaccinations. Finally, they discuss the positive role of probiotic bacteria and immunomodulatory metabolites in modifying the gut microbiome and metabolomics in order to improve the protective effects of vaccinations.

It is a well-written manuscript based on an extensive bibliography .They provide an in-depth discussion and conclusion of the subject that will be of high interest to the scientific community.

My suggestion is TO ACCEPT and publish this paper in its present form.

Author Response

We thank you for your encouraging comments.